# The Role of β-Core Fragment hCG in Embryo Implantation and Early Pregnancy

**DOI:** 10.3390/ijms26167974

**Published:** 2025-08-18

**Authors:** Ji Soo Ryu, Nu Ri Yang, Yu Ha Shim, Yu Jin Kim, Won Jae Kwag, Jin Dong Chang, Jae Ho Lee

**Affiliations:** 1Department of Biomedical Science, CHA University, Pocheon 13488, Gyeonggi-do, Republic of Korea; jisuho5229@gmail.com (J.S.R.); yuhaa525@gmail.com (Y.H.S.); 2CHA Fertility Center Seoul Station, Seoul 04637, Republic of Korea; ynr1122@chamc.co.kr (N.R.Y.); yj_kim@chamc.co.kr (Y.J.K.); 3Department of Obstetrics and Gynecology, Graduate School of Medicine, CHA University, Pocheon 13488, Gyeonggi-do, Republic of Korea; 4R&D Center, ADTech Co. Ltd., Gunpo-si 15845, Gyeonggi-do, Republic of Korea; wj_Kwag@adtchip.com (W.J.K.); jd_Chang@adtchip.com (J.D.C.)

**Keywords:** β-core fragment hCG, placenta, endometrium, proliferation, pregnancy

## Abstract

Human chorionic gonadotropin (hCG) is a pregnancy biomarker, and five forms of this hormone are involved in female physiological regulation. β-core fragment hCG (bcf-hCG) is a fragment of hCG whose biological role in female reproduction has not been completely elucidated. This study aimed to investigate its role in embryo implantation and maintenance of a pregnancy-supportive environment. We analyzed the protein expression pattern of bcf-hCG in the intrauterine environment during early pregnancy by performing western blotting and immunohistochemistry with a monoclonal anti-bcf-hCG antibody. We performed a cell proliferation assay in the presence of bcf-hCG compared with intact hCG. We conducted an ex vivo study by performing intrauterine injection of bcf-hCG or intact hCG in mice. Endometrial thickness was measured using histological methods, and uterine gene and protein expression were analyzed following intrauterine injection of bcf-hCG. We evaluated the effect of bcf-hCG on embryo implantation in the uterus. bcf-hCG was highly abundant in the placenta and epithelial stromal glands of the uterine endometrium during early pregnancy and significantly induced proliferation of a stromal epithelial cell line. Intrauterine injection of bcf-hCG induced expression of specific genes and proteins, including homeobox A10, for embryo implantation and placental development. Upon embryo transfer, the implantation rate of bcf-hCG-treated embryos was higher than that of control embryos. In conclusion, bcf-hCG plays a role in the proliferation of glandular epithelial cells in the endometrium and placenta during early pregnancy. Therefore, bcf-hCG is an early-phase pregnancy biomarker that maintains the initial phase of pregnancy.

## 1. Introduction

Human chorionic gonadotropin (hCG) is a hormone synthesized predominantly by syncytiotrophoblastic cells of the placenta during gestation [1]. It induces the corpus luteum to secrete progesterone, thereby sustaining pregnancy [2,3]. Small quantities of hCG are also synthesized in the pituitary gland, liver, and colon [4]. Certain neoplasms can also secrete hCG or hCG-related hormones. Trophoblastic neoplasms (hydatidiform moles, choriocarcinoma, and germ cell tumors) are associated with elevated serum levels of hCG-related molecules. hCG is a glycoprotein composed of two subunits, α and β. Multiple forms are present in serum and urine during pregnancy, including intact hCG and each free subunit [5]. hCG is primarily metabolized by the liver, and approximately 20% is excreted in urine. The β subunit is degraded in the kidneys to form a core fragment, which is measured by urine hCG assays [6].

hCG plays a pivotal role in early pregnancy and is utilized for various applications in reproductive medicine. It belongs to the glycoprotein hormone family, which also includes follicle-stimulating hormone, luteinizing hormone, and thyroid-stimulating hormone [7,8]. hCG is expressed in the pituitary and placenta and is involved in several functions during pregnancy, including maintaining corpus luteal progesterone production, promoting growth and differentiation of the uterus, placenta, and fetus [9,10], and inhibiting macrophage-mediated destruction of fetal-placental components [11]. There are several forms of hCG, including hCG-h, which is the most prevalent variant found in malignant trophoblastic disease and exhibits 100% sensitivity and specificity in distinguishing malignant and pre-malignant conditions [12,13]. Due to its prolonged circulating half-life of 36 h, hCG is useful for early pregnancy testing and monitoring. Although rodents, including mice, do not endogenously produce human chorionic gonadotropin (hCG), they express LH receptors that can respond to exogenously administered hCG. In this study, exogenous hCG was injected into mice to evaluate its effect on uterine and placental development. In this study we injected hCG into mice, and then we analyzed the effect and role of hCG in the female reproductive organs.

β-core fragment hCG (bcf-hCG) is a biologically inactive, low-molecular-weight molecule comprising the core portion of the β subunit [14]. It appears to be a major degradation product of the metabolism of hCG and its β subunit and is found in large quantities in the urine of pregnant women and some patients with malignancies [15,16]. Smaller quantities are found in the blood and urine of non-pregnant individuals and patients with benign disorders [14]. There is no report regarding the role of bcf-hCG in embryo implantation and pregnancy. Bcf-hCG is a specific isoform of hCG that is released into the bloodstream and urine during pregnancy [17]. bcf-hCG is a sensitive and specific marker of pregnancy and is used in various clinical settings to diagnose and monitor pregnancy. However, its biological activity has not been clearly elucidated from early pregnancy to fetal maintenance. Therefore, this study aimed to investigate the role of bcf-hCG in embryo implantation into the uterus and uterine development for pregnancy maintenance.

## 2. Results

### 2.1. Characterization of bcf-hCG During Early Pregnancy

Immunohistochemistry was performed to determine the localization of bcf-hCG in female reproductive organs and fetal tissues during early and mid-gestation. As bcf-hCG is a metabolite derived from exogenously administered whole hCG in this study, the observed signal reflects exogenous bcf-hCG. bcf-hCG localized predominantly to the placenta (Figure 1A–D). Epithelial stromal glands exhibited strong immunoreactivity for bcf-hCG (Figure 1E,F). In addition, the corpus luteum of the ovaries was positively stained for bcf-hCG, particularly during the luteal phase, indicating that bcf-hCG plays a role in luteal function (Figure 1G,H). To quantitatively assess bcf-hCG expression, Western blot analysis was conducted of various female reproductive tissues. bcf-hCG expression was higher in the placenta than in the uterus, ovaries, and fetal tissue (Figure 2A,B).

These findings suggest that bcf-hCG plays a distinct role during early pregnancy, particularly at implantation sites. The high expression of bcf-hCG in the placenta and fetal tissues implies it is involved in embryo implantation and early placental development, which are distinct from the functions of other forms of hCG.

### 2.2. Effect of Intrauterine Injection of bcf-hCG

Next, we investigated the role of bcf-hCG in the uterus. Direct intrauterine injection of bcf-hCG was performed, and histological analysis was conducted. bcf-hCG-treated uteri contained markedly more uterine glands within the endometrial stroma than control and intact hCG-treated uteri (Figure 3A). Glands were counted in histological sections of uteri. Control, bcf-hCG-treated, and intact hCG-treated uteri contained 41.5 ± 1.5 (*n* = 3), 82.5 ± 2.0 (*n* = 3), and 37.5 ± 2.5 (*n* = 3) glands, respectively (Figure 3B). The endometrium was significantly thicker in bcf-hCG-treated uteri than in control uteri (Figure 3C). By contrast, treatment with intact hCG did not significantly change endometrial thickness compared with the control.

These findings suggest that bcf-hCG promotes endometrial thickening more effectively than intact hCG and thereby contributes to a marked increase in the number of uterine glands, which are essential for secretion of implantation-supporting factors such as Hoxa10.

### 2.3. bcf-hCG Increases Cell Proliferation and Proliferation Marker Expression in the Mouse Uterus

We treated primary mouse endometrial stromal cells with DMEM/F12 supplemented with 10% fetal bovine serum and 30 pmol/mL bcf-hCG or intact hCG. After 72 h, microscopic images demonstrated that the cell density was higher in the bcf-hCG-treated group than in the control (Figure 4A). This visual observation was quantitatively supported by a significant increase in the cell number in the bcf-hCG-treated group compared with the control (Figure 4B). Consistently, mRNA expression of Ki67, a cell proliferation marker, was significantly higher in the bcf-hCG-treated group than in the control and intact hCG-treated groups (Figure 4C).

### 2.4. bcf-hCG Increases Expression of Decidualization and Placental Development Markers in the Mouse Uterus

We compared expression of decidualization and placental development markers between control, bcf-hCG-treated, and intact hCG-treated uteri. Real-time qPCR revealed that mRNA expression of Hoxa10, Hoxa11, Cyclin D3, Cdk4, Egr, Pigf, and Hand2 was significantly higher in bcf-hCG-treated uteri than in control and/or intact hCG-treated uteri (Figure 5A–J). Expression of Dedd and Cdk6 did not significantly differ between the control and experimental groups and Msx1 which showed a stepwise increase from the control to the bcf-hCG and intact hCG-treated groups. Immunohistochemical analysis demonstrated that expression of PR and Hoxa10 was markedly higher in bcf-hCG-treated uteri than in control and intact hCG-treated uteri. The intensity and distribution of brown staining were markedly enhanced in the bcf-hCG-treated group, indicative of the robust and widespread expression of PR and Hoxa10 throughout uterine tissue (Figure 6A). By contrast, minimal staining was observed in control uteri, while intact hCG-treated uteri exhibited intermediate expression levels, indicative of the relatively attenuated activation of these proteins. Protein expression of PR and Hoxa10 was markedly higher in bcf-hCG-treated uteri than in control and intact hCG-treated uteri (Figure 6B). Two distinct bands corresponding to PR-B (~116 kDa) and PR-A (~81 kDa) were detected in all groups. PR-A expression was markedly higher in bcf-hCG-treated uteri than in control and intact hCG-treated uteri.

### 2.5. Implantation Rates of bcf-hCG-Treated Embryos

Control, bcf-hCG-treated, and intact hCG-treated embryos were transferred into pseudopregnant mice. Control embryos displayed insufficient competence for implantation, resulting in a low implantation rate. By contrast, intact hCG- and bcf-hCG-treated embryos had significantly higher implantation rates (Figure 7A,B). The implantation rates of bcf-hCG- and intact hCG-treated embryos did not significantly differ, indicating that these treatments promote successful embryo implantation.

## 3. Discussion

This study provides significant insights into the role of bcf-hCG in early pregnancy and its potential implications for reproductive biology. The characterization of bcf-hCG in early pregnancy demonstrates its significant role in reproductive processes, particularly implantation and early placental development. Immunohistochemistry showed that bcf-hCG predominantly localized to the placenta and epithelial stromal glands, indicating that its function is critical in these tissues. The strong expression of bcf-hCG in ovaries during the luteal phase suggests it plays a role in luteal function, which is essential to maintain pregnancy. While previous research primarily characterized bcf-hCG as a metabolic byproduct or diagnostic biomarker of pregnancy and certain malignancies [15], this study presents novel evidence that it plays a physiological role in embryo implantation and early placental development. bcf-hCG is also involved in the regulation of fetal growth and development. It is released by the placenta in response to the needs of the developing fetus and plays a role in the maintenance of the placental barrier, which is crucial for fetal survival. In addition to its role in pregnancy, bcf-hCG has also been implicated in the development of certain diseases and conditions. For example, elevated levels of bcf-hCG are associated with pre-eclampsia, a serious complication of pregnancy that can lead to organ damage and even death. Overall, bcf-hCG is an important hormone that plays a crucial role in the regulation of pregnancy and fetal development.

We first investigated the expression profile of bcf-hCG in the female reproductive system. Immunohistochemistry demonstrated that bcf-hCG predominantly localized to the placenta and fetal tissues during early pregnancy. The strong expression of bcf-hCG in epithelial stromal glands and the corpus luteum suggests it plays a crucial role in luteal function and early placental development. Western blotting further confirmed that bcf-hCG expression was higher in the placenta than in other reproductive tissues, indicating it is involved in embryo implantation and placental development.

Next, we studied the role of bcf-hCG in the uterus and endometrium during early pregnancy and the underlying mechanism. Intrauterine injection of bcf-hCG significantly increased the number of uterine glands in the endometrial stroma. This finding is particularly important because uterine glands are essential to supply vital factors such as Hoxa10, which is required for successful embryo implantation [18,19]. The marked increase in the number of glands in bcf-hCG-treated uteri compared with control and intact hCG-treated uteri underscores the critical role of bcf-hCG in enhancing uterine receptivity and supporting early pregnancy. The proliferation assay indicated that bcf-hCG enhances proliferation of primary mouse endometrial stromal cells. The increases in absorbance upon treatment with low concentrations of bcf-hCG suggest that bcf-hCG stimulates cell proliferation, although this effect was not dose dependent. The enhanced cell proliferation observed upon bcf-hCG treatment compared with the control suggests that bcf-hCG promotes cellular growth and development, which may contribute to placental development and successful pregnancy outcomes. Real-time qPCR analysis revealed that bcf-hCG significantly upregulated expression of key genes involved in uterine receptivity and cell proliferation, including Hoxa10, Hoxa11, Cyclin D3, Cdk4, Egr, Pigf, and Hand2 [20,21,22,23,24,25,26]. These genes play pivotal roles in endometrial preparation for embryo implantation and placental development. The lack of significant changes in Dedd and Cdk6 expression suggests pathway-specific actions of bcf-hCG [27]. Msx1 plays a crucial role in regulating uterine receptivity and promoting successful embryo implantation. It acts as a molecular switch, influencing the transition of the uterine lining from a non-receptive to a receptive state, essential for embryo attachment and invasion [28]. This qPCR data was observed following intrauterine administration of bcf-hCG. suggesting that bcf-hCG may exert its effects through mechanisms independent of the LH/hCG receptor. Based on the PCR data, it represents that while the luteinizing hormone (LH) is not a direct regulator of HOXA10 and HOXA11, it plays a key upstream role. LH triggers ovulation and, critically, stimulates the formation of the corpus luteum, which is the main source of progesterone in the ovary [29]. Further studies are warranted to elucidate the precise molecular mechanisms underlying these effects.

We also evaluated the effect of bcf-hCG during the early phase of implantation and fetal development by performing ex vivo experiments. This finding emphasizes the importance of bcf-hCG for enhancing the uterine environment to support pregnancy. Importantly, embryo transfer experiments confirmed that the implantation rate was significantly higher in the bcf-hCG-treated group than in the control group, suggesting that bcf-hCG improves embryo viability and implantation competence. While both intact hCG and bcf-hCG significantly enhanced the implantation rate, the latter promoted glandular proliferation and endometrial gene expression more than the former. This difference likely reflects structural or functional distinctions between these isoforms, which potentially confer differences in receptor binding affinity or intracellular signaling dynamics. Overall, these findings provide robust evidence that bcf-hCG plays a distinct and essential role during early pregnancy by promoting uterine gland proliferation, modulating gene expression, and enhancing cell proliferation at implantation sites and during early placental development. Its elevated expression in the placenta and fetal tissues, coupled with its capacity to improve uterine receptivity, underscores its potential as a critical regulator of reproductive success. Further investigations are warranted to elucidate the underlying molecular mechanisms and to evaluate the potential therapeutic applications of bcf-hCG in reproductive medicine.

Despite these compelling findings, several limitations must be acknowledged. First, this study was conducted in a murine model; hence, direct extrapolation of these results to human reproductive physiology and clinical scenarios should be approached with caution. While our findings are promising, validation in human endometrial tissues or clinical studies will be critical for confirming translational relevance and therapeutic applicability. Second, the specific receptors and downstream signaling mechanisms that mediate the effects of bcf-hCG remain unclear, necessitating further mechanistic investigations. Additionally, potential long-term effects of bcf-hCG treatment on placental function and fetal development remain to be elucidated in longitudinal studies. Future research should focus on identifying the precise receptor interactions and molecular pathways activated by bcf-hCG. Translational studies involving human endometrial tissues, alongside clinical trials incorporating patients with recurrent implantation failure or thin endometrial linings, could validate the therapeutic potential of bcf-hCG. Furthermore, detailed comparative analyses of structural-functional relationships between intact hCG and bcf-hCG are essential to comprehensively delineate their biological roles.

## 4. Materials and Methods

### 4.1. Animals and Uterine Injection of hCG Samples

BDF1 mice (C57BL/6 × DBA/2; F1) were obtained from Orient Bio Co., Ltd. (Seoul, The Republic of Korea). All animal experiments, breeding, and care procedures were performed following the regulations of the Institutional Animal Care and Use Committee (IACUC) of CHA University. IACUC approval (approval number IACUC240049) was obtained before initiation of the study. Each experiment used nine mice (control = 3; test = 6). BDF1 female mice were injected with pregnant mare serum gonadotropin (RP1782725000; BioVendor, Brno, Czech Republic), followed by hCG (668900221; LG Chem, Seoul, Republic of Korea) 48 h later to induce estrous synchronization. Aligned with the estrous cycle, all groups, including the control, were pre-treated with pregnant mare serum gonadotropin (PMSG; RP1782725000; BioVendor, Brno, Czech Republic) to synchronize the estrous phase, and the control (no treatment), bcf-hCG, and highly purified intact hCG were prepared at 30 pmol/mL and injected directly into the uterus. mNSET™ (60010; Paratechs, Lexington, KY, USA) was used for administration. Uterine tissue samples were collected at 24 and 72 h post-injection.

### 4.2. Cell Culture and Proliferation Assay

The optimal concentration of bcf-hCG and intact hCG was determined for treatment of primary mouse endometrial stromal cells. Cells were seeded at a density of 3 × 10^3^ cells (100 μL) per well in a 96-well plate (90 × 20 mm; SPL-20100; SPL, Pocheon, The Republic of Korea). The next day, cells were treated with 30 pmol/mL bcf-hCG or intact hCG overnight, and then cell proliferation was assessed using a WST-8 Cell Viability Assay Kit (QM2500; BIOMAX, Seoul, The Republic of Korea). This assay assessed the cell number by measuring the reaction between WST-8 and electrons released from NADH, enabling calculation of the proliferation rate. In total, 10 μL of WST-8 solution was added to the cell-containing medium at 24, 48, and 72 h, according to the manufacturer’s protocol. After allowing the reaction to proceed for 4 h, absorbance at 450 nm was measured using a microplate spectrometer (Epoch™ Microplate Spectrophotometer; BioTek, Winooski, VT, USA).

### 4.3. Isolation of Endometrial Stromal Cells

Primary mouse endometrial stromal cells were isolated from the uteri of BDF1 mice. Uteri were collected, minced into small pieces, and digested with 0.25% trypsin (25200072; Gibco, Grand Island, NY, USA) at 37 °C for 1 h. Following digestion, the samples were vigorously vortexed in Hanks’ Balanced Salt Solution (LB203-04; Welgene, Seoul, The Republic of Korea) to release stromal cell sheets. The suspension was centrifuged at 500× *g* for 5 min, and the cell pellet was resuspended in DMEM/F-12 (LM002-05, Welgene) supplemented with 10% fetal bovine serum (16000-044; Gibco, Grand Island, CA, USA) and 1% penicillin-streptomycin (15140122; Gibco, Grand Island, CA, USA). The cell suspension was then passed through a 40 μm nylon mesh filter (93040, SPL) to remove undigested tissue fragments. Stromal cells were seeded in DMEM/F-12 (LM002-05, Welgene), incubated overnight, and grown to 70–90% confluency.

### 4.4. Histological Analysis

Uteri of control mice (*n* = 3), bcf-hCG-treated mice (*n* = 3), and intact hCG-treated mice (*n* = 3) were fixed with 4% paraformaldehyde for 30 min at room temperature, dehydrated in an ethanol series (70%, 80%, 95%, and 100% ethanol), embedded in paraffin, and stained with hematoxylin and eosin using routine protocols. The tenth and sixteenth cross-sections of each uterine sample were stained with hematoxylin and eosin to quantify the number of endometrial stromal glands and to assess endometrial thickness. Images were acquired using an inverted light microscope (Eclipse Ti2; Nikon, Tokyo, Japan) equipped with a camera (DS-Ri2, Nikon) and imaging software (NIS-Elements ver. 4.4., Nikon).

### 4.5. Development of Anti-bcf-hCG Monoclonal Antibody

Immunogen as bcf-hCG was isolated and purified to high purity from pregnant women’s urine using immuno-affinity chromatography. Next, we immunized the ICR mice with purified bcf-hCG following routine immunization protocol. A bcf-hCG-specific mouse monoclonal antibody (clone: 2G12) was coupled to NHS-activated Sepharose 4 Fast Flow (Cytiva Cat. No. 17090601) resin and used as an immuno-affinity column. Then we generated hybridomas using the cell fusion method after immunizing mice with crude hCG. Subsequently, we selected specific antibodies using bcf-hCG (NIBSC code: 99/708, South Mimms, UK) as the standard antibody, which was purchased from NIBSC (National Institute for Biological Standards and Control). The specificity of the selected anti-bcf-hCG monoclonal antibody (Mab) was confirmed by ELISA (Appendix A). We tested its reactivity against standard samples purchased from NIBSC, including bcf-hCG, native hCG, beta-free hCG, and alpha hCG. The results confirmed that the antibody specifically binds only to the beta core fragment of hCG. The detailed ELISA procedure and binding profiles are presented in Appendix A.

### 4.6. Immunohistochemistry

Tissue-containing paraffin sections on glass slides were deparaffinized by incubation in a xylene series ranging from 100% to 75% for 10 min each and then rehydrated by incubation in an ethanol series ranging from 100% to 75% for 10 min each at room temperature. For antigen retrieval, the tissue sections were boiled in 0.01 M citrate buffer (pH 6.0) for 20 min in an autoclavable jar. After permeabilization with phosphate-buffered saline (PBS) containing 1% Triton X-100 for 1 h at room temperature, the sections were washed three times with fresh PBS. Subsequently, the sections were incubated overnight at 4 °C with anti-homeobox A10 (Hoxa10; sc-271428; Santa Cruz, Dallas, TX, USA), anti-progesterone receptor (PR; MA1-410; Invitrogen, Waltham, MA, USA), and anti-bcf-hCG (ADTech, Gunpo, The Republic of Korea) antibodies diluted 1:100 in PBS containing 0.03% bovine serum albumin. For DAB staining, a streptavidin–horseradish peroxidase (HRP) system included in the detection kit (ab64264, Abcam, Cambridge, UK) was used to apply the secondary antibody, and counterstaining with hematoxylin was performed for 5 min at room temperature. Then, slides were mounted using mounting media and overlaid with coverslips. Images of DAB staining were acquired using an inverted light microscope (Eclipse Ti2, Nikon) equipped with a camera (DS-Ri2, Nikon) and imaging software (NIS-Elements ver. 4.4., Nikon).

### 4.7. Western Blotting

Proteins were extracted with Pro-Prep protein lysis buffer (17081; iNtRON, Seongnam, The Republic of Korea) and quantified with a protein quantification assay kit (BCA0500, BIOMAX). Samples were then boiled in 4× Laemmli sample buffer (BR1610747; Bio-Rad, Hercules, CA, USA) containing 2-mercaptoethanol (1610710, Bio-Rad), loaded into an 8% sodium dodecyl sulphate polyacrylamide gel, and electrophoresed at 60 V for 30 min followed by 120 V for 1 h. Proteins were transblotted onto a nitrocellulose membrane (Bio-Rad) at 400 mA for 90 min. The membrane was incubated with blocking buffer (TBS-T containing 5% bovine serum albumin) for 1 h at room temperature, followed by a solution containing the following primary antibodies overnight at 4 °C: anti-PR (MA1-410, Invitrogen), anti-HOXA10 (sc-271428, Santa Cruz), and anti-β-actin (MA5-15739; Thermo Fisher Life Technologies, Waltham, MA, USA). The blotted membrane was incubated with an anti-mouse secondary antibody (A21127, Invitrogen) for 1 h at room temperature. Immunoreactive bands were detected using enhanced chemiluminescence detection reagent (Clarity™ Western blot substrate, Bio-Rad, Hercules, CA, USA). Images of the bands were acquired using ImageSaver Version 6 (ATTO, Tokyo, Japan). The intensity of each band was analyzed with CA4 analyzer software (ATTO, Tokyo, Japan; version 3.0.1). The experiment was repeated three times under the same conditions with different samples.

### 4.8. Real-Time Quantitative PCR (qPCR)

Total RNA was extracted using TRIzol according to the manufacturer’s procedures (15596026, Invitrogen). The RNA concentration was quantified using a microplate spectrometer (Epoch™ Microplate Spectrophotometer, BioTek) and adjusted to 100 ng/μL. RNA was reverse transcribed into cDNA with RT PreMix (dT20) (Bioneer, Seoul, The Republic of Korea) using a SimpliAmp Thermal Cycler (Life Technologies, Carlsbad, CA, USA). Real-time qPCR was performed with a real-time PCR machine (CFX Connect, Bio-Rad), SYBR Green Supermix (Bio-Rad), and primers specific to mouse homeobox A10 (Hoxa10), homeobox A11 (Hoxa11), death effector domain-containing DNA-binding protein (Dedd), Cyclin D3, cyclin-dependent kinase 4/6 (Cdk4/6), early growth response (Egr), placental growth factor (Pigf), heart and neural crest derivatives expressed 2 (Hand2), Msh Homeobox (Msx1), and ribosomal protein L7 (Rpl7). All experiments were repeated three times for statistical analysis. The sequences of primers used in this study are listed in Appendix A.

### 4.9. Assessment of Implantation After Embryo Transfer

The implantation capacities of embryos treated with bcf-hCG and intact hCG were determined using BDF1 mice. Recipient mice were prepared for pseudopregnancy. Vasectomized male and surrogate female white ICR mice were used for embryo transfer studies. Female mice were mated with male mice overnight, and sperm plugs were checked the following day. Female mice with confirmed sperm plugs were used for embryo transfer. 2PN stage were collected from BDF1 females and cultured in EmbryoMax^®^ KSOM Mouse Embryo Media (1X), Liquid, with 1/2 Amino Acids & Phenol Red (MR-121-D; Sigma-Aldrich, Saint Louis, MO, USA) until they reached the morula stage. These embryos were further cultured to the blastocyst stage by day 3 in vitro. Blastocysts on culture day 3 were implanted via direct intrauterine transfer using mNSET™ (60010, Paratechs). After embryo transfer, implantation rates were compared between the control and experimental groups.

### 4.10. Statistical Analysis

All data are expressed as the mean ± standard error of the mean (SEM) of triplicate measurements. Statistical analyses were performed with a one-way analysis of variance (ANOVA) and Bonferroni’s test of variance using GraphPad Prism software (version 9). For datasets that did not meet the assumption of normality, non-parametric Kruskal–Wallis tests were applied. The significance level was set at * *p* < 0.05, ** *p* < 0.01, or *** *p* < 0.001.

## 5. Conclusions

In conclusion, this study significantly advances current understanding of hCG biology by identifying previously unrecognized physiological roles of bcf-hCG in embryo implantation and early placental development. Our results suggest that bcf-hCG is a promising therapeutic target or adjunctive treatment to enhance endometrial receptivity and improve outcomes of assisted reproductive technologies. Further detailed research is warranted to confirm the relevance and clinical utility of bcf-hCG for human fertility treatments.

## Figures and Tables

**Figure 1 ijms-26-07974-f001:**
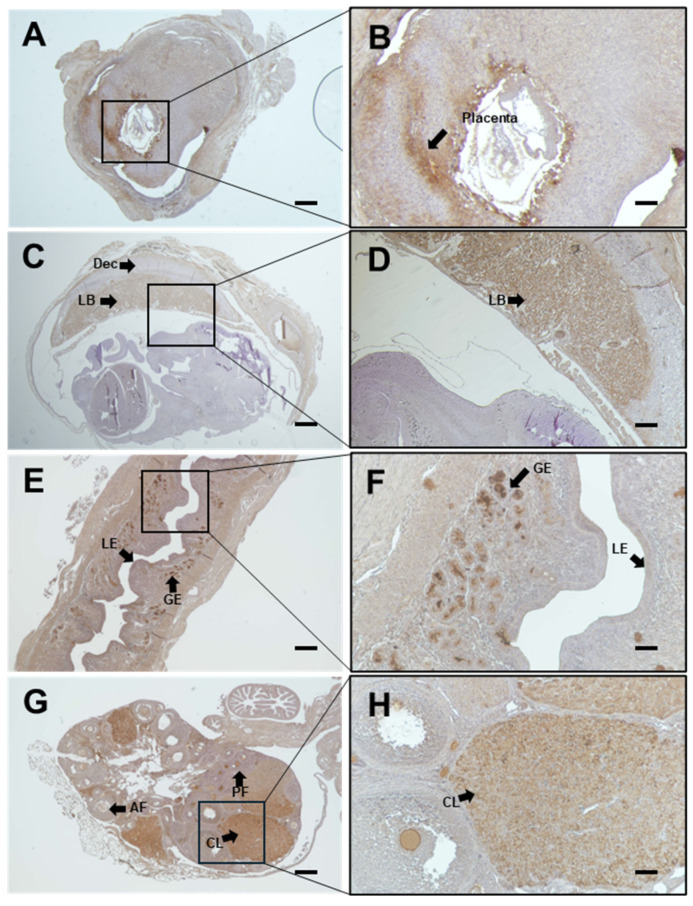
Immunohistochemical analysis of β-core fragment hCG (bcf-hCG) expression in fetal tissue, placenta, uterus, and ovary. (**A**,**B**) Placenta with embryonic day 7 fetus (black arrows indicate the placenta); (**C**,**D**) Placenta with embryonic day 12 fetus, showing Dec (decidua) and LB (labyrinth zone); (**E**,**F**) Early pregnancy uterus; black arrows indicate the epithelial stromal glands**,** LE (luminal epithelium), and GE (glandular epithelium); (**G**,**H**) Early pregnancy ovary showing AF (antral follicle), PF (primordial follicle), and CL (corpus luteum), respectively, at different magnifications (right: 100×, left: 40×). Scale bars are 200 and 100 μm on the right and left, respectively (*n* = 3).

**Figure 2 ijms-26-07974-f002:**
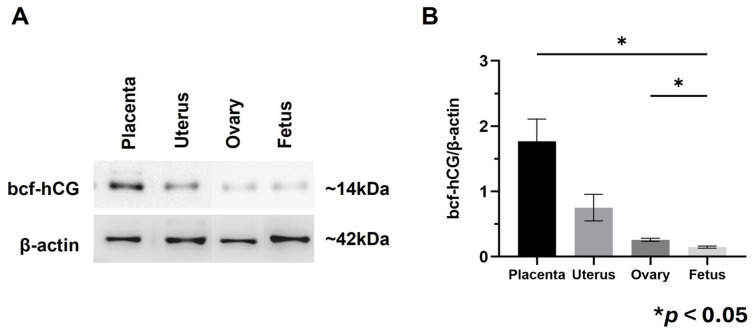
Protein expression of bcf-hCG and intact hCG in fetal tissue, placenta, uterus, and ovary. (**A**) Western blot analysis of bcf-hCG expression. (**B**) Quantification of normalized band intensities for bcf-hCG. The data are presented as means ± SEM of three replicates. Significant differences are indicated by asterisks (* *p* < 0.05).

**Figure 3 ijms-26-07974-f003:**
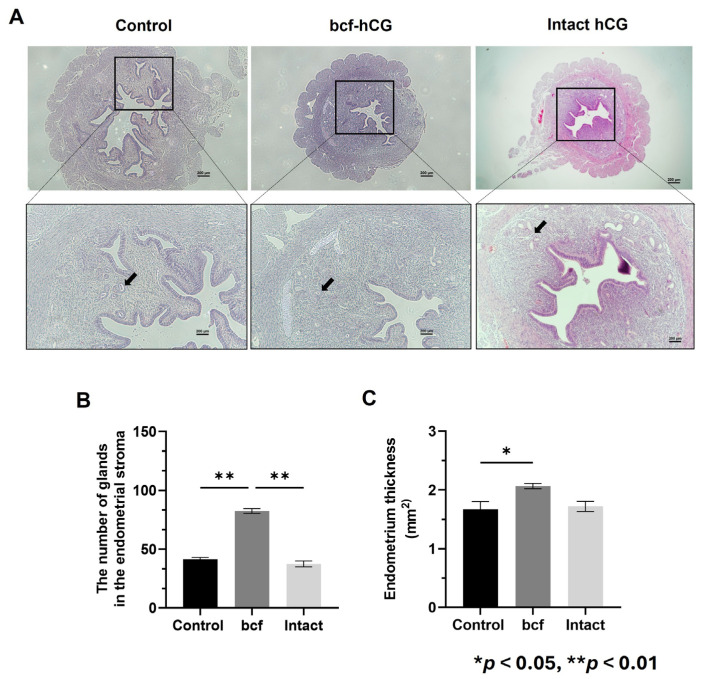
Characterization of bcf-hCG-treated uteri. (**A**) Hematoxylin and eosin-stained control, bcf-hCG-treated, and intact hCG-treated mouse uteri. Black arrows indicate endometrial glands. Images are shown at different magnifications (top: 40×, bottom: 100×). Scale bar = 200 μm. (**B**) Number of glands in the endometrial stroma. (**C**) Endometrial thickness. Data are presented as mean ± SEM of three replicates. Significant differences are indicated by asterisks (* *p* < 0.05 and ** *p* < 0.01). Control uteri (*n* = 3), bcf-hCG-treated uteri (*n* = 3), and intact hCG-treated uteri (*n* = 3).

**Figure 4 ijms-26-07974-f004:**
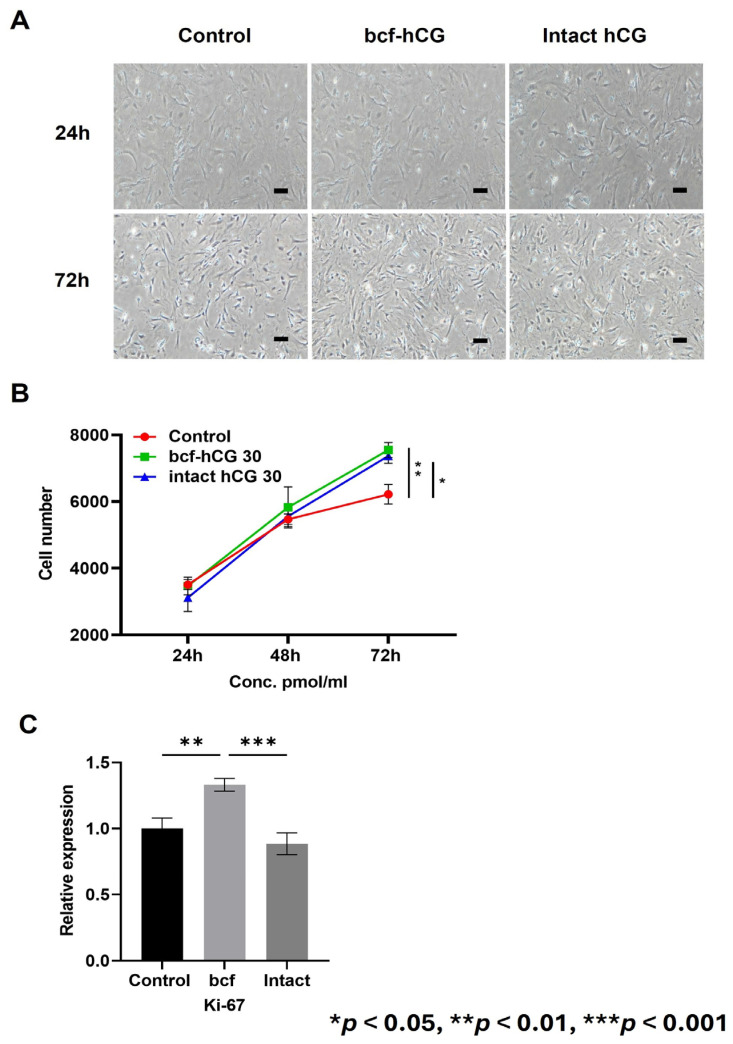
Proliferation of primary mouse endometrial stromal cells treated with bcf-hCG and intact hCG. (**A**) Representative microscopic images of control mouse endometrial stromal cells and those treated with intact hCG and bcf-hCG (30 pmol/mL) for 24 and 72 h. Scale bar is 200 μm. (**B**) Cell proliferation after treatment with bcf-hCG and intact hCG for 24, 48, and 72 h. (**C**) Real-time qPCR analysis of Ki-67 expression normalized to β-actin expression. Data are presented as mean ± SEM of three replicates. Significant differences are indicated by asterisks (* *p* < 0.05, ** *p* < 0.01, and *** *p* < 0.001). Statistical significance was determined by a one-way ANOVA.

**Figure 5 ijms-26-07974-f005:**
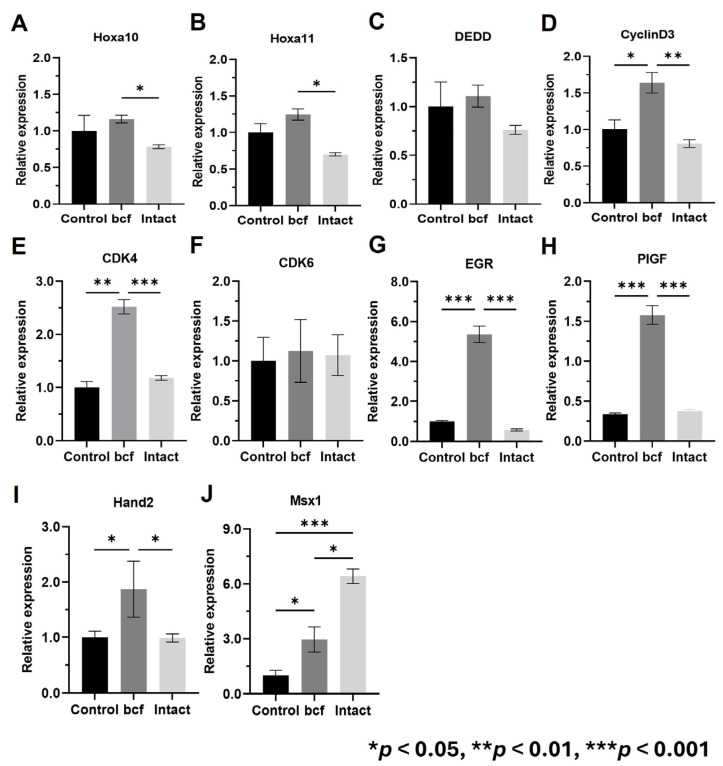
Expression of decidualization and early pregnancy maintenance genes in uteri treated with bcf-hCG and intact hCG. (**A**) Hoxa10, (**B**) Hoxa11, (**C**) Dedd, (**D**) Cyclin D3, (**E**) Cdk4, (**F**) Cdk6, (**G**) Egr, (**H**) Pigf, (**I**) Hand2, and (**J**) Msx1. Data are presented as mean ± SEM of three replicates. Significant differences are indicated by asterisks (* *p* < 0.05, ** *p* < 0.01, and *** *p* < 0.001). Statistical significance was determined by a one-way ANOVA.

**Figure 6 ijms-26-07974-f006:**
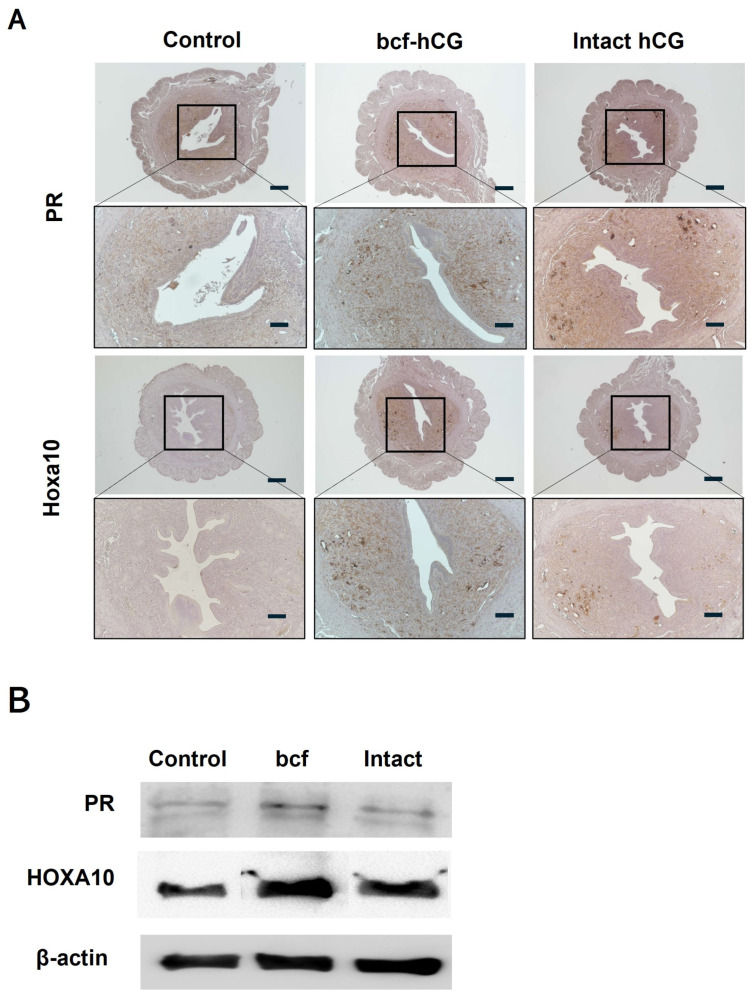
Expression of Hoxa10 and PR in mouse uteri treated with bcf-hCG and intact hCG. (**A**) Immunohistochemical DAB staining of Hoxa10 and PR. Images are shown at different magnifications (top: 40×, bottom: 100×). Scale bars are 200 μm. (**B**) Proteins were extracted from uterine tissues and analyzed by western blotting using antibodies against Hoxa10 and PR.

**Figure 7 ijms-26-07974-f007:**
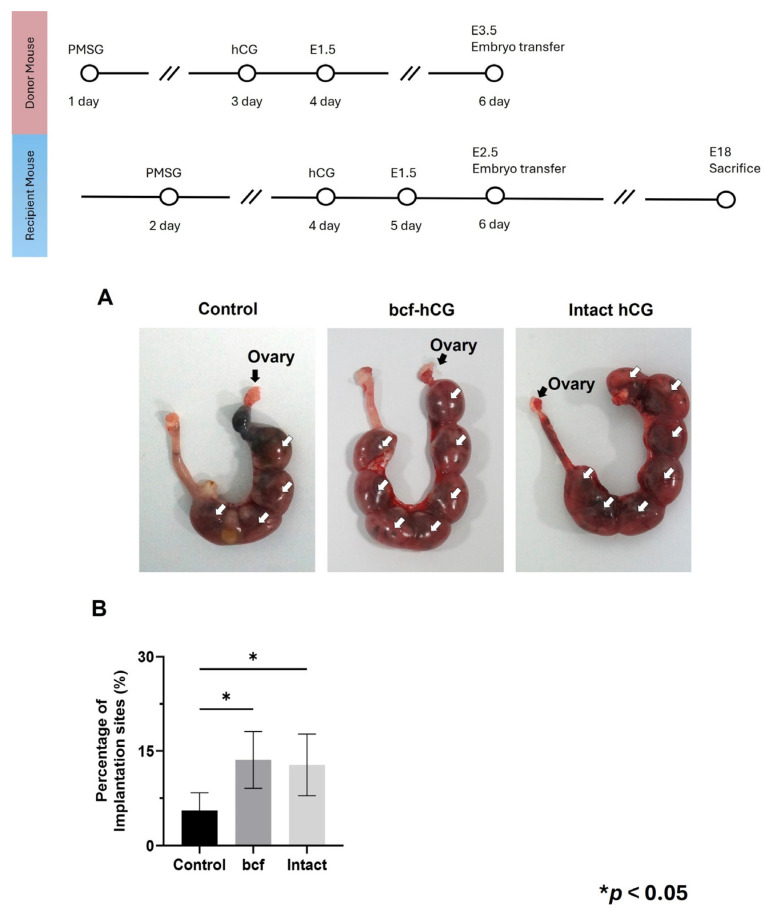
Implantation of embryos treated with bcf-hCG and intact hCG following transfer into recipient mice. (**A**) Representative images showing implantation sites. (**B**) Implantation rates of control, bcf-hCG-treated, and intact hCG-treated embryos. Data are presented as mean ± SEM of three replicates. Significant differences are indicated by asterisks (* *p* < 0.05). Statistical significance was determined by a one-way ANOVA.

## Data Availability

The data presented in this study are available in the article.

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
