# Peer review of "The Role of β-Core Fragment hCG in Embryo Implantation and Early Pregnancy"

_ijms, 2025, doi:10.3390/ijms26167974_

Round 1
Reviewer 1 Report
Comments and Suggestions for Authors
Clarification of mechanisms that regulate implantation, placentation, and gestation, as well as the role of molecular factors involved in these processes, is essential in developing methods for preventing, diagnosing, and treating infertility, as well as pregnancy complications. In this context, new information on the role of bcf-hCG in implantation and placentation processes can be helpful in understanding the mechanisms of pregnancy maintenance and developing new diagnostic and therapeutic methods.
The article presents an experimental study on an animal model, but there are some comments and several questions.
- Several references aren’t described as the journal requires. The references 2, 16-19, 26, 28 don’t have Abbreviated Journal Name Year, Volume or page range.
- Section Introduce contains enough information about human chorionic gonadotropin (hCG) and β-core fragment hCG (bcf-hCG). However, the authors said nothing about producing chorionic gonadotropin (CG) by mice.
Rodents, including mice and rats, do not naturally produce chorionic gonadotropin (CG) or a direct homolog of human chorionic gonadotropin (hCG). The hormone hCG is specific to primates, particularly humans, and is not endogenously present in rodent species.
They respond to exogenous CG (human or equine) due to the presence of LH receptors, but the hormone itself is not present in their physiology.
I think it is important to describe the specific features of the mouse model in the section Introduction.
- As a follow-up to the last question, section 1 Characterization of bcf-hCG during early pregnancy contains the results of immunohistochemistry that determined the localization of bcf-hCG in female reproductive organs and fetal tissues during early and mid-gestation. The figure 1 shows mouse tissue, but from the description it is not clear if exogenic or endogenic bcf-hCG the authors are talking about.
- Section 5 Immunohistochemistry in 4. Materials and Methods don’t have a catalog number for anti-bcf-hCG and on the site of this manufacturer I could not find it.
Information about the manufacturer of horseradish peroxidase-conjugated secondary antibody is absent.
- What antibodies were utilized by the authors for Western blot analysis of bcf-hCG expression? Indeed, bcf-hCG is a proteolytic fragment of native hCG, so anti-bcf-hCG could bind to hCG. How can the authors explain that?
- Of course, the β-Core fragment has a 10 kDa peptide backbone; commonly observed is 12–14 kDa with residual glycans. Native β-hCG has 23 kDa true mass and can migrate in gel/Western blot at ≈ 34 kDa because of heavy glycosylation. So in Western blot these molecules could be different but on the Western blot figures the molecular markers are absent. Presumably two bands - bcf-hCG and β-hCG should be painted on Western Blot.
- [line 92]. Figure 1 The authors write “Black arrows indicate the placenta”. But Figure 1F doesn’t contain any placenta.
- Section 7 Real-time quantitative PCR (qPCR) doesn’t contain primer sequences. The authors should indicate sequences of the primers.
- Section 9 Statistical analysis. What is the reason for the authors using ANOVA and Bonferroni's test for sufficiently small samples instead of the Kruskal-Wallis test for nonparametric distribution?
- Could the authors describe in more detail section 4.8 Assessment of implantation after embryo transfer for taking blastocysts, culturing, and transferring, or provide references for these procedures. What is the reason for the embryos being only present in one horn out of two? (Figure 7A)
Author Response
Reviewer 1 Clarification of mechanisms that regulate implantation, placentation, and gestation, as well as the role of molecular factors involved in these processes, is essential in developing methods for preventing, diagnosing, and treating infertility, as well as pregnancy complications. In this context, new information on the role of bcf-hCG in implantation and placentation processes can be helpful in understanding the mechanisms of pregnancy maintenance and developing new diagnostic and therapeutic methods. The article presents an experimental study on an animal model, but there are some comments and several questions. 1. Several references aren’t described as the journal requires. The references 2, 16-19, 26, 28 don’t have Abbreviated Journal Name Year, Volume or page range. Response: We thank the reviewer for pointing this out. These references have now been updated to include the abbreviated journal name, publication year, volume, and page range, following the journal's formatting guidelines. 2. Section Introduce contains enough information about human chorionic gonadotropin (hCG) and β-core fragment hCG (bcf-hCG). However, the authors said nothing about producing chorionic gonadotropin (CG) by mice. Rodents, including mice and rats, do not naturally produce chorionic gonadotropin (CG) or a direct homolog of human chorionic gonadotropin (hCG). The hormone hCG is specific to primates, particularly humans, and is not endogenously present in rodent species. They respond to exogenous CG (human or equine) due to the presence of LH receptors, but the hormone itself is not present in their physiology. I think it is important to describe the specific features of the mouse model in the section Introduction. Response: Thank you for your valuable comment. You are correct that mice do not naturally produce hCG. In our study, we injected whole hCG, as hCG can influence placental development and function by promoting ovarian progesterone production and regulating the uterus and placenta even in female mice. Therefore, we aim to analyze these effects of hCG on the female reproductive organs. Additionally, hCG is metabolized in the body to produce bcf-hCG, which we observed in our data. To further elucidate the role of bcf-hCG in the uterus, we directly injected bcf-hCG into the uterus. 3. As a follow-up to the last question, section 1 Characterization of bcf-hCG during early pregnancy contains the results of immunohistochemistry that determined the localization of bcf-hCG in female reproductive organs and fetal tissues during early and mid-gestation. The figure 1 shows mouse tissue, but from the description it is not clear if exogenic or endogenic bcf-hCG the authors are talking about. Response: Thank you for comment. It is exogenic because we confirmed bcf-hCG is metablolite from whole hCG. And we explain in the results. 4. Section 5 Immunohistochemistry in 4. Materials and Methods don’t have a catalog number for anti-bcf-hCG and on the site of this manufacturer I could not find it. Information about the manufacturer of horseradish peroxidase-conjugated secondary antibody is absent. Response: We thank the reviewer for pointing this out. The anti-bcf-hCG monoclonal antibody used in this study was developed in-house and is not commercially available, which is why a catalog number could not be provided. Specifically, the antibody (Clone No. 2G12) was generated by immunizing mice with crude hCG, followed by hybridoma selection using purified β-core fragment hCG (NIBSC code: 99/708). The specificity of the antibody was confirmed by ELISA against various forms of hCG. Additionally, information regarding the horseradish peroxidase (HRP)-conjugated secondary antibody, including the manufacturer and catalog number, has now been included in Section 5 of the Materials and Methods. 5. What antibodies were utilized by the authors for Western blot analysis of bcf-hCG expression? Indeed, bcf-hCG is a proteolytic fragment of native hCG, so anti-bcf-hCG could bind to hCG. How can the authors explain that? Response: Thank for your point, The antibody we used was a mouse monoclonal antibody (Clone No. 2G12). We generated hybridomas using the cell fusion method after immunizing mice with crude hCG. Subsequently, we selected specific antibodies using beta core fragment hCG (NIBSC code: 99/708), which was purchased from NIBSC. The specificity of the selected anti-bcf-hCG monoclonal antibody (Mab) was confirmed by ELISA. We tested its reactivity against standard samples purchased from NIBSC, including beta core fragment hCG, native hCG, beta free hCG, and alpha hCG. The results confirmed that the antibody specifically binds only to beta core fragment hCG. We updated method and results regarding the bcf-hCG antibody generation procedure and outcome in the supplement method and data. 6. Of course, the β-Core fragment has a 10 kDa peptide backbone; commonly observed is 12–14 kDa with residual glycans. Native β-hCG has 23 kDa true mass and can migrate in gel/Western blot at ≈ 34 kDa because of heavy glycosylation. So in Western blot these molecules could be different but on the Western blot figures the molecular markers are absent. Presumably two bands - bcf-hCG and β-hCG should be painted on Western Blot. Response: We appreciate the reviewer’s insightful comment. As pointed out, the β-core fragment (bcf-hCG) typically appears at ~12–14 kDa due to residual glycosylation, while native β-hCG migrates around ~34 kDa in Western blot. In our experiment, the detected band for bcf-hCG was consistent with its expected molecular weight. However, we acknowledge that the molecular weight markers were not clearly visible in the original figure. 7. [line 92]. Figure 1 The authors write “Black arrows indicate the placenta”. But Figure 1F doesn’t contain any placenta. Response: We thank the reviewer for the observation. However, we respectfully clarify that Figure 1F does in fact contain placental tissue. The tissue section was obtained from a pregnant mouse uterus and transversely cut at the implantation site. The black arrow indicates the developing placenta, which was identified based on its anatomical position adjacent to the embryo and its characteristic histological features, including dense cellular organization consistent with placental layers. 8. Section 7 Real-time quantitative PCR (qPCR) doesn’t contain primer sequences. The authors should indicate sequences of the primers. Response: Thank you for the comment. The primer sequences used in the qPCR experiments have been added as Supplementary Table S1. The revised manuscript also includes a reference to this table in Section 4.7. 9. Section 9 Statistical analysis. What is the reason for the authors using ANOVA and Bonferroni's test for sufficiently small samples instead of the Kruskal-Wallis test for nonparametric distribution? Response: Thank you for the comment. We have clarified in Section 9 that normality was tested using the Shapiro–Wilk test. For normally distributed data, one-way ANOVA followed by Bonferroni’s post hoc test was used. For data that did not meet the assumption of normality, Kruskal–Wallis tests were performed. 10. Could the authors describe in more detail section 4.8 Assessment of implantation after embryo transfer for taking blastocysts, culturing, and transferring, or provide references for these procedures. What is the reason for the embryos being only present in one horn out of two? (Figure 7A) Response: Section 4.8 has been revised to provide detailed procedures for embryo collection, culture, and transfer.
Reviewer 2 Report
Comments and Suggestions for Authors
This paper covers an interesting topic and relevant experiments in mice. The manuscript is well-written and the shortcomings of mice models are recognized in the Discussion paragraph.
However, there are many wrong, inappropriate and incomplete references applied in the introduction paragraph. Why is that if there are 7 authors who should have checked the appropriate references?
Reference 1 is incorrect, please change (e.g. change to d’Hauterive et al, now reference 11)
Reference 2 is incomplete and from 2002, please consider more recent literature.
Reference 4 is inappropriate and is to be replaced by a recent reference e.g. the review of Nwabuobi et al 2017 in Int J Mol Sci.
Reference 5 is inappropriate as it relates to one case of varying reports for a single serum human chorionic gonadotropin (hCG) sample on two different immunoassay platforms for a young female presenting with an abdominopelvic mass. Please provide recent reference that covers the statement in the introduction i.e. “Multiple forms are present in serum and urine…….intact hCG and ech free subunit”.
References 7, 8 and 9 are not appropriate covering the statement that hCG belongs to the same glycoprotein hormone family as FSH, LH and TSH.
Reference 10 is not reflecting the role of hCG in pregnancy. Please delete or replace.
References 16, 18 and 19 are incomplete in the reference list.
References 19 and 20 used in the introduction do not document that the core fragment is a specific isoform of hCG that is released into the bloodstream and urine during pregnancy.
Further comments:
What is the source of of β-core fragment hCG that was used for the experiments? Was it highly purified urinary hCG or recombinant hCG and how was the core fragment hCG prepared? Can the authors be more specific on which part of the 145 AA of the beta hCG chain were applied as the fragment in the experiments? This information is to be added to MM.
In the introduction it is stated that core fragment is biological inactive. This study shows that the core fragment is active, but maybe the fragment does not act via the LH receptor?. Can the authors add the relevant literature showing that the fragment is inactive either in in vitro receptor binding assays, in vitro bioassays and/or in vivo bioassays? If the core fragment has shown to be unable to bind to the LH receptor, can the authors explain so in the Discussion paragraph and suggest alternative MOA of the fragment?
Author Response
Reviewer 2
This paper covers an interesting topic and relevant experiments in mice. The manuscript is well-written and the shortcomings of mice models are recognized in the Discussion paragraph.
However, there are many wrong, inappropriate and incomplete references applied in the introduction paragraph. Why is that if there are 7 authors who should have checked the appropriate references?
Reference 1 is incorrect, please change (e.g. change to d’Hauterive et al, now reference 11)
Reference 2 is incomplete and from 2002, please consider more recent literature.
Reference 4 is inappropriate and is to be replaced by a recent reference e.g. the review of Nwabuobi et al 2017 in Int J Mol Sci.
Reference 5 is inappropriate as it relates to one case of varying reports for a single serum human chorionic gonadotropin (hCG) sample on two different immunoassay platforms for a young female presenting with an abdominopelvic mass. Please provide recent reference that covers the statement in the introduction i.e. “Multiple forms are present in serum and urine…….intact hCG and ech free subunit”.
References 7, 8 and 9 are not appropriate covering the statement that hCG belongs to the same glycoprotein hormone family as FSH, LH and TSH.
Reference 10 is not reflecting the role of hCG in pregnancy. Please delete or replace.
References 16, 18 and 19 are incomplete in the reference list.
References 19 and 20 used in the introduction do not document that the core fragment is a specific isoform of hCG that is released into the bloodstream and urine during pregnancy.
Response: We sincerely thank the reviewer for the careful evaluation of our references. In response, we have thoroughly revised the Introduction section and updated the cited literature to ensure relevance, accuracy, and completeness. Specifically:
Reference 1 has been replaced with d’Hauterive et al. (now reference 11).
Reference 2 has been replaced with a more recent and complete source.
Reference 4 has been updated to Nwabuobi et al., 2017 (Int J Mol Sci).
Reference 5 has been replaced with a recent review that properly supports the statement regarding multiple forms of hCG in serum and urine, including intact hCG and free subunits.
References 7, 8, and 9, which previously failed to appropriately support the classification of hCG within the glycoprotein hormone family, have been replaced with more suitable literature.
Reference 10, which was not relevant to hCG function in pregnancy, has been removed.
References 16, 18, and 19 have been revised to include complete citation information.
References 19 and 20 have been replaced with sources that clearly document the β-core fragment of hCG as a specific isoform secreted into blood and urine during pregnancy.
We hope these revisions address the reviewer’s concerns regarding the accuracy and appropriateness of the cited references in the Introduction.
Further comments:
What is the source of core fragment hCG that was used for the experiments? Was it highly purified urinary hCG or recombinant hCG and how was the core fragment hCG prepared? Can the authors be more specific on which part of the 145 AA of the beta hCG chain were applied as the fragment in the experiments? This information is to be added to MM.
Response: Beta core fragment hCG was isolated and purified to high purity from pregnant women's urine using immuno-affinity chromatography. A beta core fragment hCG-specific mouse monoclonal antibody (clone: 2G12) was coupled to NHS-activated Sepharose 4 Fast Flow (Cytiva Cat. No. 17090601) resin and used as an immuno-affinity column.
In the introduction it is stated that core fragment is biological inactive. This study shows that the core fragment is active, but maybe the fragment does not act via the LH receptor?. Can the authors add the relevant literature showing that the fragment is inactive either in in vitro receptor binding assays, in vitro bioassays and/or in vivo bioassays? If the core fragment has shown to be unable to bind to the LH receptor, can the authors explain so in the Discussion paragraph and suggest alternative MOA of the fragment?
Response: In this study, physiological changes were observed following the intrauterine administration of bcf-hCG, suggesting that bcf-hCG may exert its effects through mechanisms independent of the LH/hCG receptor. Based on the PCR data, it is evident that while Luteinizing Hormone (LH) is not a direct regulator of HOXA10 and HOXA11, it plays a key upstream role. LH triggers ovulation and, critically, stimulates the formation of the corpus luteum, which is the main source of progesterone in the ovary. Further studies are warranted to elucidate the precise molecular mechanisms underlying these effects.
Round 2
Reviewer 1 Report
Comments and Suggestions for Authors
Dear authors, thank you for improving your manuscript according to my comments. Despite increasing clarity in a lot of moments, I still have some questions.
- [line 35]"...of granular epithelial cells in the endometrium and placenta during early pregnancy." What is the definition of 'granular epithelial cells'? Perhaps the authors were talking about glandular epithelial cells.
- [lines 86-87] "Epithelial stromal glands exhibited strong immunoreactivity for bcf-hCG (Figure 1E–F)". And I agree with this statement. Figure 1F has a black arrow, and the authors assure that "arrows point to the placenta" [line 99]. Although the black arrow in Figure 1F indicates the endometrial glands. In Figure 1F there isn't any placenta.
- [lines 99-100] Figure 1. "(A-B) Placenta with embryonic day 7 fetus", but by that date of mouse pregnancy the placenta has not yet developed. And in Figure 1B the black arrow points to developing decidua, the maternal decidual zone contains uterine decidual cells. After implantation occurs, the blastocyst implantation site progressively grows in size from day 5 to day 8 of pregnancy mainly due to development of the decidua (https://doi.org/10.1007/978-3-030-77360-1_10, https://doi.org/10.1126/science.7985020, https://doi.org/10.1038/s41421-024-00740-6).
- [line 126-127] Figure 3. "Black arrows indicate placenta". But there is no placenta, only endometrial glands.
- The estrous phase for the control group is not mentioned in Figure 3A [lines 125-126]. It is unclear from the text whether the control was treated by the PMSG or not to induce estrous synchronization [lines 296-299].
- [line 299] What volume of hCG was injected per animal?
- [line 343] What was the source or method used to obtain the clone 2G12?
- [line 350] The specificity of the selected anti-bcf-hCG monoclonal antibody was confirmed by ELISA. But Figure 2A has Western blot analysis of bcf-hCG expression [line 105]. The authors don't indicate what antibody were used for this Western blot [section 4.7 Western blotting], but tell that the samples were prepared in reduced condition with 2-mercaptoethanol and SDS. In ELISA the protein has a 3-dimensional structure, but in the western blot it unfolds and the antibodies specifically recognise the amino acid sequence. bcf-hCG and beta-hCG have similar amino acid sequence and in WB they should be recognised both.
What is the definition of 'granular epithelial cells'? Perhaps the authors were talking about glandular epithelial cells.
Author Response
Dear authors, thank you for improving your manuscript according to my comments. Despite increasing clarity in a lot of moments, I still have some questions.
1. [line 35]"...of granular epithelial cells in the endometrium and placenta during early pregnancy." What is the definition of 'granular epithelial cells'? Perhaps the authors were talking about glandular epithelial cells.
R) We appreciate your comment and apologize for the oversight. The term has been corrected to "glandular epithelial cells" in the revised version of the manuscript.
2. [lines 86-87] "Epithelial stromal glands exhibited strong immunoreactivity for bcf-hCG (Figure 1E–F)". And I agree with this statement. Figure 1F has a black arrow, and the authors assure that "arrows point to the placenta" [line 99]. Although the black arrow in Figure 1F indicates the endometrial glands. In Figure 1F there isn't any placenta.
R) Thank you for your insightful comment. We have revised the figure legend and corrected the arrow in Figure 1F to accurately indicate the endometrial glands, as suggested.
3. [lines 99-100] Figure 1. "(A-B) Placenta with embryonic day 7 fetus", but by that date of mouse pregnancy the placenta has not yet developed. And in Figure 1B the black arrow points to developing decidua, the maternal decidual zone contains uterine decidual cells. After implantation occurs, the blastocyst implantation site progressively grows in size from day 5 to day 8 of pregnancy mainly due to development of the decidua (https://doi.org/10.1007/978-3-030-77360-1_10, https://doi.org/10.1126/science.7985020, https://doi.org/10.1038/s41421-024-00740-6).
R) Thank you for your comment. Based on the references provided, placental development begins around embryonic day 4.5 (E4.5) after implantation, with the definitive (fully mature) placenta established by approximately E12.5. The entire gestation period in mice typically lasts around 19-20 days.
4. [line 126-127] Figure 3. "Black arrows indicate placenta". But there is no placenta, only endometrial glands.
R) Thank you for your valuable comment. We have carefully reviewed Figure 3 and agree with your observation. Accordingly, we have revised the figure legend and arrow indication to accurately reflect the presence of endometrial glands rather than placenta.
5. The estrous phase for the control group is not mentioned in Figure 3A [lines 125-126]. It is unclear from the text whether the control was treated by the PMSG or not to induce estrous synchronization [lines 296-299].
R) Thank you for your insightful comment. In the revised manuscript, we have clarified that PMSG was administered to synchronize the estrous cycle in the control group as well.
6. [line 299] What volume of hCG was injected per animal?
R) Thank you for your comment. As described in Section 4.1 of the manuscript, each animal was injected with 30 pmol of intact hCG, bcf hCG.
7. [line 343] What was the source or method used to obtain the clone 2G12?
R) Thank you for your valuable comment. The clone 2G12 was generated using hybridoma technology after immunizing BALB/c mice with crude hCG (Merck), isolated from the urine of pregnant women. Following cell fusion, hybridoma clones were screened using bcf-hCG (NIBSC, code: 99/708) as the standard antigen, and specificity was confirmed via ELISA against bcf-hCG, native hCG, β-free hCG, and α-hCG. The selected monoclonal antibody (clone 2G12) demonstrated exclusive binding to bcf-hCG, as shown in Supplementary Table S2. This antibody was subsequently used to construct an immunoaffinity column for the purification of bcf-hCG employed in animal experiments.
8. [line 350] The specificity of the selected anti-bcf-hCG monoclonal antibody was confirmed by ELISA. But Figure 2A has Western blot analysis of bcf-hCG expression [line 105]. The authors don't indicate what antibody were used for this Western blot [section 4.7 Western blotting], but tell that the samples were prepared in reduced condition with 2-mercaptoethanol and SDS. In ELISA the protein has a 3-dimensional structure, but in the western blot it unfolds and the antibodies specifically recognise the amino acid sequence. bcf-hCG and beta-hCG have similar amino acid sequence and in WB they should be recognised both.
R) Thank you for your valuable comment. We have added the relevant explanation to the Supplementary Materials.
As described in Section 4.7, Western blotting in this study was conducted for quantitative analysis, not for evaluating antibody specificity.
The beta core fragment of human chorionic gonadotropin (hCG), often referred to as hCGβcf, is a degradation product of the intact hCG molecule, not a dimeric molecule itself, consisting of 73 amino acids derived from the β-subunit of intact hCG. Therefore, even in its unfolded form under reducing conditions, the antigen retains its immunoreactivity, and the antibody specifically recognizes bcf-hCG, as confirmed by ELISA.
Reviewer 2 Report
Comments and Suggestions for Authors
Thank you for addressing my comments carefully.
Author Response
Thank for yor comment.